# DNA-binding of the Tet-transactivator curtails antigen-induced lymphocyte activation in mice

Eleonora Ottina [1,2,6], Victor Peperzak[2,3,7], Katia Schoeler[1], Emma Carrington[2], Roswitha Sgonc[4], Marc Pellegrini[2,3], Simon Preston[2,3], Marco J. Herold[2,3], Andreas Strasser[2,3] & Andreas Villunger[1,5]

The Tet-On/Off system for conditional transgene expression constitutes state-of-the-art technology to study gene function by facilitating inducible expression in a timed and reversible manner. Several studies documented the suitability and versatility of this system to trace lymphocyte fate and to conditionally express oncogenes or silence tumour suppressor genes in vivo. Here, we show that expression of the tetracycline/doxycycline-controlled Tet-transactivator, while tolerated well during development and in immunologically unchallenged animals, impairs the expansion of antigen-stimulated T and B cells and thereby curtails adaptive immune responses in vivo. Transactivator-mediated cytotoxicity depends on DNA binding, but can be overcome by BCL2 overexpression, suggesting that apoptosis induction upon lymphocyte activation limits cellular and humoral immune responses. Our findings suggest a possible system-intrinsic biological bias of the Tet-On/Off system in vivo that will favour the outgrowth of apoptosis resistant clones, thus possibly confounding data published using such systems.

[1] Division of Developmental Immunology, Biocenter, Medical University of Innsbruck, Innrain 80, 6020 Innsbruck, Austria. [2] The Walter and Eliza Hall Institute for Medical Research, 1G Royal Parade, Victoria 3052, Australia. [3] Department of Medical Biology, The University of Melbourne, Parkville, Victoria 3050, Australia. [4] Division of Experimental Pathophysiology and Immunology, Biocenter, Medical University of Innsbruck, Innrain 80, 6020 Innsbruck, Austria. [5] The Tyrolean Cancer Research Institute, Innrain 66, 6020 Innsbruck, Austria. [6]Present address: Retroviral Immunology, The Francis Crick Institute, 1 Midland Road, London NW1 1AT, UK. [7]Present address: Laboratory of Translational Immunology (LTI), Utrecht (UMCU), Heidelberglaan 100, 3584 CX Utrecht, The Netherlands. Correspondence and requests for materials should be addressed to E.O. (email: Eleonora.Ottina@crick.ac.uk) or to A.V. (email: andreas.villunger@i-med.ac.at)

Genetically modified mice are an important tool for the investigation of gene function in health and disease. Traditionally, the function of a gene is explored by manipulation of its expression levels either by deletion or over-expression of its wild-type coding DNA sequence or a mutated form. Conversely, disruption or subtle modifications of the endogenous gene locus are achieved by homologous recombination in embryonic stem cells used to generate gene-modified mice[1, 2]. Loss-of-function studies have expanded our knowledge about any given gene, on the basis of the analysis of the phenotype(s) that result from its modification or ablation. However, phenotypes may often be confounded by functional overlap between several genes within the same family[3, 4] or by ill-defined compensatory events enabling the development of a functional organism around a potentially detrimental null-allele or destructive random transgene insertion[5–8]. As a result, constitutive loss-of-function phenotypes often do not mimic the consequences of acutely induced gene ablation in the adult organism in its entirety or in a given tissue of interest. Often it is preferable to gain spatial and temporal control over gene deletion or overexpression. Site-specific DNA-recombinase systems, such as the Cre-loxP system, were developed to meet this need and enable integration, deletion or inversion of an endogenous or integrated DNA fragment in a controlled manner[9].

Even though CRE-mediated recombination facilitates cell type-specific and timed ablation of conditional alleles, as well as the controlled activation of introduced transgenes[10], and the advent of CRISPR/Cas9 technology has even made simultaneous targeting of multiple genes in vivo applicable[11, 12], all these approaches still have limitations. Most importantly, the often artificially high-transgene expression levels may cause toxicity to some cell types, and promiscuous binding to, and cleavage of, genomic DNA by the CRE recombinase can be fatal[13–16]. Similar limitations may apply to the Cas9 endonuclease that can stochastically bind many coding gene loci[17]. Hence, phenotypes noted in genetically manipulated mice might not always mirror the function of any given gene in the adult or in the tissue of interest. For these reasons, systems that enable timed and graded manipulation of transgene expression or reversible gene ablation are often preferable. Hence, inducible transgene, RNA interference (RNAi) approaches are being exploited as a scalable alternative to conventional transgenic or loss-of-function approaches, even allowing genome-wide in vivo RNAi screening[18–21]. Genome-wide interrogation of gene function and screening methods using RNA-based CRISPR interference (CRISPRi) has also been developed. CRISPRi is based on an enzymatically dead Cas9 (dCas9) fused to a Krüppel-associated box (KRAB) transcriptional repression domain, which does not cleave the target gene, but reduces its expression when dCas9 is targeted to a transcriptional start site and inhibits transcription[22, 23]. However, promising, at the moment, the design of functional guide RNA for CRISPRi has proven to be challenging[24]; therefore, RNAi screening remains the valid method for reversible gene regulation.

To date, the most commonly used model for timed and spatial regulation of transgene/RNAi expression in mice is the *Escherichia coli*-derived tetracycline-regulated Tet-On/Off system, originally developed by Gossen and Bujard[25–27], whereas other systems, such as the lac repressor-lac operon-based transgene expression system seem less robust in vivo[28, 29]. The Tet system relies on two components, a tetracycline/doxycycline-responsive transactivator or repressor, tTA or rtTA, and the desired gene of interest placed under the control of a tetracycline-responsive promoter element (TRE). The first generation of the tTA hybrid transcription factor is a fusion protein of the prokaryotic Tet repressor, TetR, conferring DNA-binding capacity that is abrogated upon binding by tetracycline (Tet-Off), with the acidic transactivation domain of the herpes simplex virus-encoded transcription factor VP16 driving transgene expression[25]. In 1995, a complementary system based on a 'reverse' transactivator (rtTA) was developed. The rtTA differs from the tTA by four amino acids within the TetR sequence that result in a reversal of responsiveness, enabling DNA binding and induction of transgene expression upon tetracycline or doxycycline addition (Tet-On)[30]. The most commonly used version of a Tet-responsive promoter ($P_{TET}$) consists of the CMV minimal promoter, fused to seven tetracyclin-responsive operator elements, tetO, from *E. coli*, conferring DNA-binding specificity to the tTA and rtTA[27].

Despite the remarkable potential and broad applicability of the Tet system, shortcomings exist, which restrict its general use. Limitations include in particular the CMV promoter-associated leakiness and its poor expression in certain cell types[31, 32]. Hence, to overcome these limitations the $P_{TET-Tight}$ promoter system was introduced by Clontech (now Takara Bio). This promoter consists of seven tetO elements upstream of the minimal CMV promoter sequence ($-35$ to $+12$), but lacking the CMV-enhancer sequence, rendering the $P_{TET-Tight}$ transcriptionally inactive in the absence of a tetracycline/doxycycline-responsive transactivator. Alternative modifications and refinements include codon optimization and the development of the tetracycline-controlled transcriptional silencers (tTS), such as the KRAB domain of the human *Kid-1* gene. However, since the exploitation of the tTS requires the co-expression of three different transgenes, this system is mainly used for analysis in cell lines[32–34].

In mice, the Tet-On/Off system has been most widely exploited in the field of cancer research in which oncogenes, such as *MYC*, *RAS* or *BCL2*, have been overexpressed as conditional transgenes[35–38] or tumour suppressors, such as p53, PTEN or adenomatous polyposis coli (APC), were silenced by RNAi in a regulated manner. This also enabled the discovery of modifiers of oncogenesis in genome-wide in vivo RNAi screens[19, 39, 40]. Others have exploited the system to study lymphocyte development[41] or to trace lymphocyte fate by conditional expression of fluorescent markers (e.g., green fluorescent protein (GFP), mCherry)[42]. Although these studies reveal the potential of this technology, other findings aiming to investigate essential genes or genes involved in cell maintenance also point towards its limitations, as transgene silencing, as well as compensatory effects antagonizing in vivo RNAi at the protein level are apparent[43–46]. Together with a report defining the ability of tetracycline to cause mitochondrial dysfunction, and affect global gene expression profiles in mammalian cells[47], these studies suggest a system-intrinsic biological bias of the Tet system in vivo that can select for cell death-resistant clones. This hypothesis is supported and the concerns surrounding the use of this system are extended by our observation that Tet-transactivator expression impairs the survival of antigen-activated B and T cells in mice.

## Results

**Tet-transactivator expression impairs humoral immunity in mice.** During the course of our studies using transgenic mice expressing a mi-shRNA targeting the pro-survival BCL2 family member, A1[45], we observed an shRNA expression-independent effect in vav-tTA mice, where Tet-transactivator expression is limited to the haematopoietic system[48]. In steady state, animals displayed with largely unperturbed haematopoiesis and normal leukocyte subset composition in primary as well as secondary lymphatic organs (Supplementary Fig. 1). A reduction in CD11c$^+$CD8$^+$ conventional dendritic cells (cDCs) that was accompanied by a parallel increase in CD11c$^+$CD11b$^+$ cDCs was noted upon further analysis (Supplementary Fig. 2), yet, both

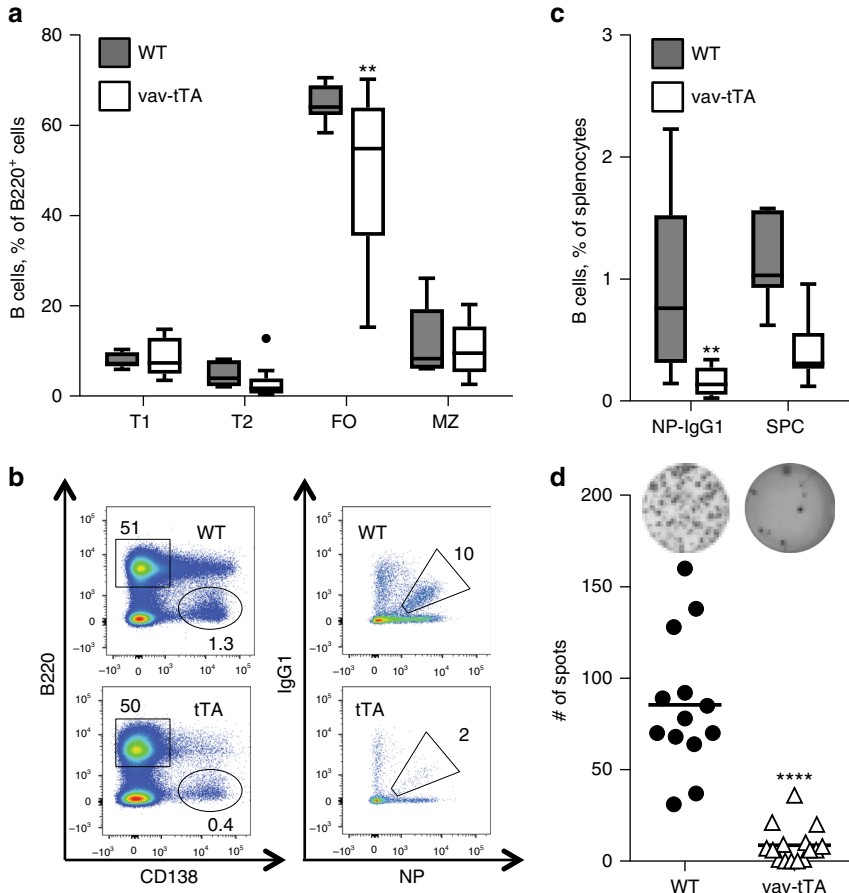

**Fig. 1** tTA transactivator expression compromises germinal centre formation and development of antigen-secreting cells. Flow cytometric analysis of splenocytes from vav-tTA mice or littermate controls at day 7 post immunization with NP-KLH. **a** Percentages of transitional type 1 (T1) B cells (B220$^{high}$ IgM$^{high}$ CD23$^{low}$ CD21$^{low}$), transitional type 2 (T2) B cells (B220$^{high}$ IgM$^{high}$ CD23$^+$ CD21$^{high}$), follicular (FO) B cells (B220$^{high}$ IgM$^{low}$ CD23$^+$ CD21$^{low}$) and marginal zone (MZ) B cells (B220$^{high}$ IgM$^{high}$ CD23$^{low}$ CD21$^{high}$). For the gating strategy, please see the Supplementary Fig. 9a. Data are presented as box-and-whiskers diagram with median and interquartile range (vav-tTA $n = 10$; littermate controls $n = 7$). **$P \leq 0.01$, vav-tTA vs. wild-type (WT), one-way ANOVA with Bonferroni correction for multiple comparisons. **b** Representative staining with B220- and CD138-specific antibodies to identify plasma cells (left) or NP-specific IgG1$^+$ germinal centre B cells (left), defined as IgM$^-$ IgD$^-$ Gr1$^-$ CD138$^-$ B220$^+$ IgG1-NP$^+$. For the gating strategy, please see the Supplementary Fig. 9b. **c** Frequencies of splenic plasma cells (SPC) (B220$^{low}$ CD138$^+$) and germinal centre NP-specific isotype-switched B cells (IgM$^-$ IgD$^-$ Gr1$^-$ CD138$^-$ B220$^+$ NP$^+$ IgG1$^+$). Data are presented as box-and-whiskers diagram with median and interquartile range (vav-tTA $n = 14$; littermate controls $n = 13$). **$P \leq 0.01$, vav-tTA vs. wild-type (WT), one-way ANOVA with Bonferroni correction for multiple comparisons. **d** Frequencies of total NP-specific IgG1-secreting cells in spleen determined by ELISPOT assay. Data are presented as scatter plots with mean; each dot represents data from a single immunized mouse as a mean of assays in two replicate wells, summarizing three independent experiments ($n = 4$–$6$ per genotype and immunization). ****$P < 0.0001$, vav-tTA vs. WT, two-sample Kolmogorov–Smirnov test

subsets were still present in significant number. Remarkably, upon immunization with the T cell-dependent model antigen, NP-KLH, vav-tTA mice displayed a reduction of splenic follicular B cells (Fig. 1a) and also failed to generate Ig class-switched antigen-specific B cells (Fig. 1b, c). Subsequent ELISPOT analysis confirmed the near complete absence of antigen-specific antibody-secreting cells in their spleens (Fig. 1d).

To link the tTA transgene expression with the impaired capacity to form antigen-specific B cells, we intercrossed vav-tTA mice with animals harbouring a Tet-responsive fluorescent reporter (DT mice). To this end we made use of two mouse strains, TRE-Renilla (Ren) and TRE-Firefly, harbouring a transgene encoding a mi-shRNA targeting Ren or Firefly luciferase downstream of an *eGFP*-encoding complementary DNA that was introduced into the *Col1a* locus by homologous recombination and placed under the control of a conventional tetracycline-responsive promoter, P$_{TET}$ (Supplementary Fig. 3a)[19]. An initial comparison of both DT strains showed no major changes in the distribution of B lymphocytes in bone marrow,

spleen or lymph nodes, when compared with single-transgenic or wild-type controls, while T2 B cells were found to be mildly reduced (Supplementary Fig. 3d). Additionally, although tTA expression did not perturb thymic T cell development (Supplementary Fig. 3e, f), in line with observations by others[49–52], the frequency of splenic CD4$^+$ effector/effector memory cells was mildly reduced (Supplementary Fig. 3h).

Moreover, while we found homogenous expression of enhanced green fluorescent protein (eGFP) in splenic T and B cells, we observed variegated reporter expression in myeloid cells (Supplementary Fig. 5a–c), a finding in line with published results[29, 44]. Lymphocyte proliferation and survival was largely unaffected upon in vitro stimulation (Supplementary Fig. 4), while GFP$^+$ granulocytes were underrepresented and showed increased apoptosis ex vivo (Supplementary Fig. 5d). Taken together, this suggests that vav-tTA mice show some alteration in their haematopoietic system that correlate with transgene expression as reported by eGFP, while problems based on random transgene insertion appear negligible in this strain.

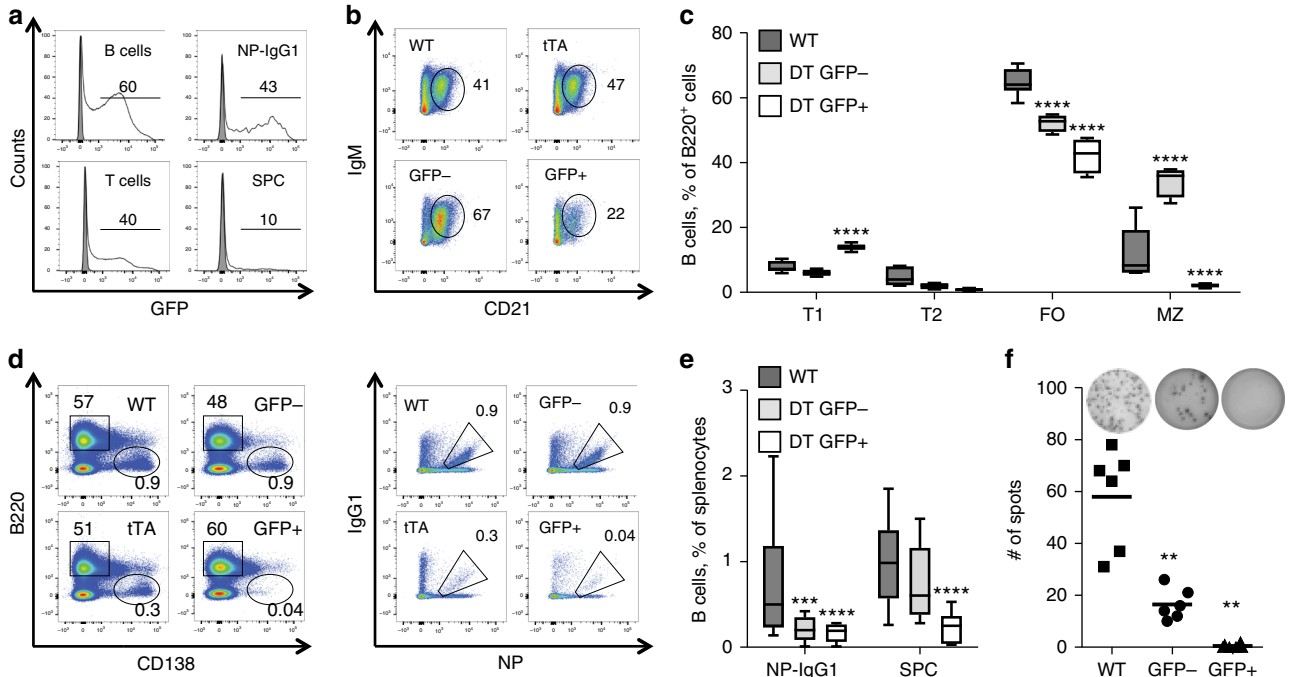

**Fig. 2** tTA expression level correlates with the severity of impairment of B cell activation. Flow cytometric analysis of splenocytes from TRE_Ren/vav-tTA or TRE-FF/vav-tTA double transgenic (DT) mice or littermate controls at day 7 after immunization with NP-KLH. **a** GFP reporter expression in B cells (B220[+]), T cells (TCRβ[+]), plasma cells (B220[low] CD138[+]) and germinal centre NP-specific B cells (IgM[−] IgD[−] Gr1[−] CD138[−] B220[+] NP[+] IgG1[+]) splenocytes of DT mice, open histograms, and wt mice, filled histograms; percentages given refer to GFP[+] cells. **b** Representative dot plots of wt, vav-tTA and DT splenocytes gated on B220[high] and CD23[low] and **c** quantification of transitional type 1 (T1) B cells (B220[high] IgM[high] CD23[low] CD21[low]), transitional type 2 (T2) B cells (B220[high] IgM[high] CD23[+] CD21[high]), follicular (FO) B cells (B220[high] IgM[low] CD23[+] CD21[low]) and marginal zone (MZ) B cells (B220[high] IgM[high] CD23[low] CD21[high]) of wt and DT mice sorted on the basis of the GFP expression level. For the gating strategy, please see the Supplementary Fig. 9a. Data are presented as box-and-whiskers diagram with median and interquartile range (DT mice n = 7; littermate controls n = 7). ***P ≤ 0.001, ****P ≤ 0.0001, GFP[+] or GFP[−] vs. controls, two-way ANOVA with Bonferroni correction for multiple comparisons. **d** Representative plasma cell staining, using B220 and CD138 specific antibodies (left) and NP-specific IgM[−] IgD[−] Gr1[−] CD138[−] B220[+] IgG1-NP[+] (right) germinal centre B cells, quantified in **e**. For the gating strategy, please see the Supplementary Fig. 9b. Data are presented as box-and-whiskers diagram with median and interquartile range (DT mice n = 9; littermate controls n = 9). **P ≤ 0.01,***P ≤ 0.001, ****P ≤ 0.0001, GFP[+] and GFP[−] vs. controls, two-way ANOVA with Bonferroni correction for multiple comparisons. **f** Frequencies of total NP-specific IgG1-secreting cells among FACS-sorted splenocytes as determined by ELISPOT assay. Each dot represents data from a single mouse as mean of assays in replicate wells from two independent experiments using three mice per genotype. Data are presented as scatter plots with mean. **P < 0.01, GFP[+] vs. GFP[−] subsets, two-sample Kolmogorov–Smirnov test

To our surprise, upon immunization with NP-KLH, reporter expression was only detected in a fraction of germinal centre (GC) B cells and almost absent in plasma cells (Fig. 2a). Furthermore, the reporter expression in activated B cells was significantly reduced when compared to mice in steady state (Fig. 2a and Supplementary Fig. 3b), suggesting counter-selection[44]. Additionally, we observed that tTA expression, as reported by GFP and confirmed via quantitative PCR (qPCR) (Supplementary Fig. 3c), correlated with a reduction of follicular B cells and a loss in marginal zone B cells (Fig. 2b,c). Moreover, tTA expression impaired the survival, differentiation and IgG1 class switch-recombination of GC B cells resulting in reduced NP-specific plasma cells (Fig. 2d–f). Toxicity clearly correlated with transgene expression levels, as GFP-negative cells, sorted from the spleen of immunized transgenic mice were still able to give rise to IgG-secreting plasma cells that could be detected by ELISPOT analysis, albeit in reduced numbers, when compared to wild-type controls (Fig. 2f). Next, we immunized a mouse strain expressing the eGFP reporter under the control of the P_TET-tight promoter (TRE_tight-Ren) that was crossed with vav-tTA mice (DTtight) (Supplementary Fig. 6a). The TRE-tight system was developed by optimization of the conventional TET promoter to reduce the leakiness and improve the levels of transgene expression. Essentially, the TRE-tight system requires higher

levels of transactivator to be present to achieve expression of the gene of interest. Consistently, the GFP-expressing cells in these DTtight mice were found to express higher level of tTA mRNA when compared to the GFP[+] cells from DT mice (Supplementary Figs. 3c and 6b). These experiments yielded highly comparable results with an almost complete loss of GC B cells and lack of NP-specific IgG-secreting plasma cells upon immunization (Supplementary Fig. 6), corroborating the findings shown in Fig. 2. Of note, also the percentages of GC B cells within the GFP-negative fraction of all double-transgenic strains we studied was reduced. This suggests that GFP-negative cells may actually represent a mixed population of cells, some of which have silenced the reporter gene expression, despite the presence of the tTA, a notion supported by our qPCR analysis (Supplementary Figs. 3c and 6b).

**Tet-transactivator expression impairs cellular immunity in mice.** As pan-haematopoietic expression of the tTA may compromise antibody production indirectly, for example, by impairing T cell help, possibly secondary to impaired DC activity, we next analysed the T cell compartment in vav-tTA mice, but were again unable to detect prominent differences in steady state (Supplementary Figs. 1, 3 and 4). Yet, upon immunization of DT

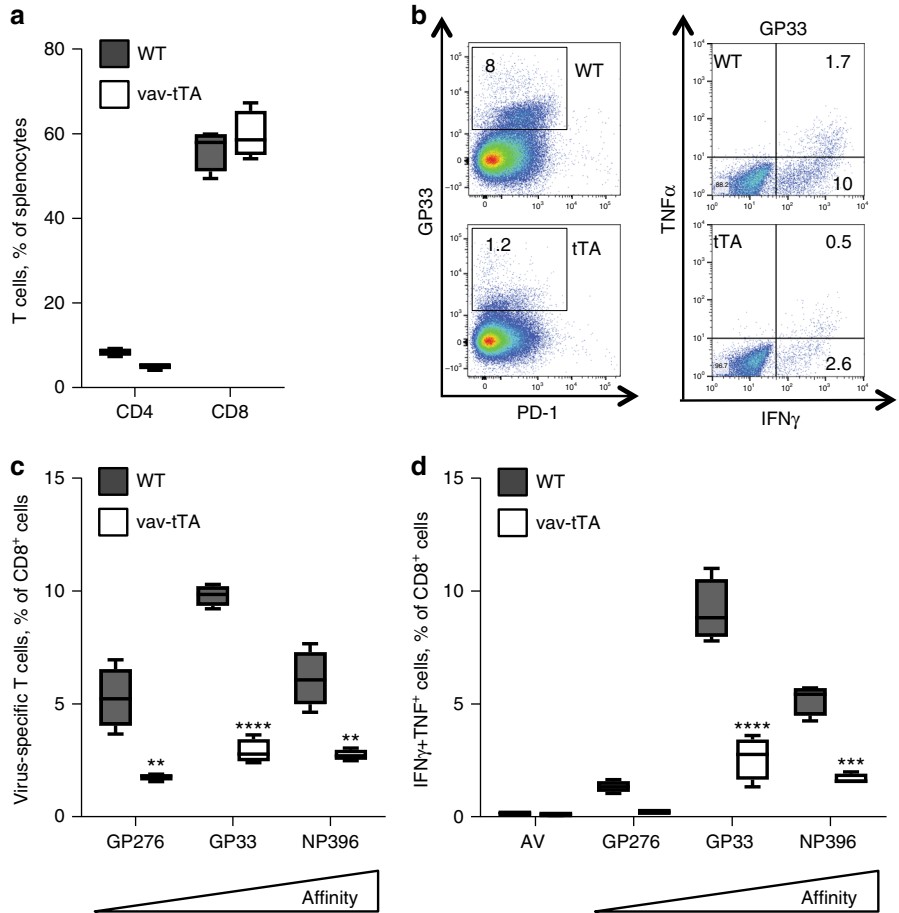

**Fig. 3** tTA transactivator expression impairs development of T cell immunity during acute viral infection. Flow cytometric analysis of splenocytes from vav-tTA mice ($n = 4$) or littermate controls ($n = 4$) at day 8 after infection with LCMV strain WE. **a** Percentages of CD4$^+$ and CD8$^+$ T cells in spleens of infected mice. **b** Flow cytometric analysis of LCMV-specific CD8$^+$ T cells in spleens of wt and vav-tTA mice upon infection. Cells were stained with antibodies against CD8, PD-1 and tetramers for low-affinity and high-affinity peptides of LCMV proteins: H-2Db/GP33 (KAVYNFATM), H-2Db/GP276 (SGVENPGGYCL) and H-2Db/NP396 (FQPQNGQFI), conjugated with streptavidin APC (left); IFNγ and TNF production in virus-specific CD8$^+$ T cells after a 4 h in vitro restimulation of splenocytes with the cognate peptide was assessed by intracellular staining and flow cytometric analysis (right). For the gating strategy, please see the Supplementary Fig. 9c, d. **c** Quantification of LCMV-reactive tetramer-positive CD8$^+$ T cells. **d** Quantification of cytokine-positive tetramer positive CD8$^+$ T cells after in vitro restimulation. Data are presented as box-and-whiskers diagram with median and interquartile range of four mice per genotype. **$P < 0.01$, ***$P \leq 0.001$, ****$P \leq 0.0001$, two-way ANOVA with Bonferroni correction for multiple comparisons

mice, we observed a clear reduction of effector T cells in the GFP$^+$ fraction, suggesting that tTA expression may also be cytotoxic to activated T cells (Supplementary Fig. 7). To test if this phenomenon might be limited to antigen-driven humoral immune responses, we monitored the T cell response of vav-tTA mice upon viral infection (Fig. 3a). To this end, we challenged tTA mice with the LCMV (lymphocytic choriomeningitis virus) WE strain, causing acute infection in mice where neither B cells nor DCs participate in the primary anti-viral response[53–56]. Tetramer staining at the peak of infection revealed a clear reduction of virus-specific CD8$^+$ T cells (Fig. 3b, c) in vav-tTA mice. Furthermore, we observed impaired cytokine production after in vitro re-stimulation of ex vivo of vav-tTA-expressing CD8 T cells using the LCMV-derived GP276, GP33 or NP396 peptides (Fig. 3d).

These findings suggest that tTA expression also curtails T cell activation and that the reduction of NP-specific B cells in the experiments described above may in part be due to impaired T cell help. To address this possibility more directly, we made use of mice that harbour the tTA transgene under control of the Ig heavy gene enhancer to target its expression selectively in B cells. Similar to vav-tTA mice, Eμ-tTA transgenic animals presented

with normal B cell numbers in the naive state (not shown), but failed to generate antigen-specific activated B cells or antibody-secreting plasma cells upon immunization (Fig. 4). This suggests that tTA expression in B cells alone is sufficient to limit their clonal expansion, Ig class switching and differentiation into plasma cells.

**Transactivator cytotoxicity depends on DNA binding**. We reasoned that the tTA-dependent toxicity in activated B cells might depend on its overexpression per se causing proteotoxic stress or off-target effects triggered upon DNA binding. This could be elicited by, for example, promiscuous recognition of endogenous DNA elements by the DNA-binding domain or, alternatively, by deregulation of other genes in transgene proximity due to the viral VP16 transactivator, leading to activation of gene expression profiles incompatible with lymphocyte activation.

To discriminate between these possibilities, we made use of mice that express the rtTA in all cells and compared B cell responses in vav-tTA with those of CAG-rtTA mice upon immunization in the presence or absence of doxycycline. Our

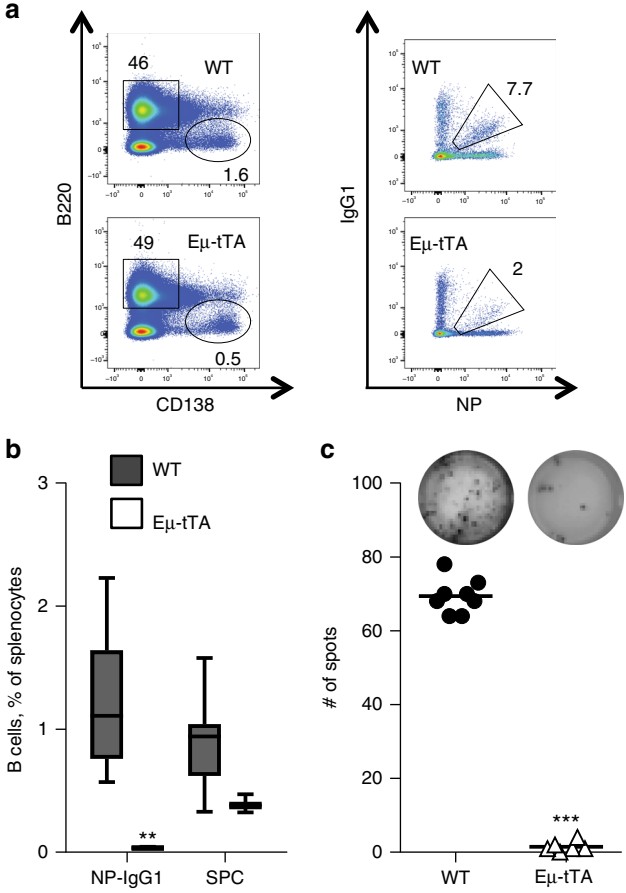

**Fig. 4** The tTA transactivator induced loss of germinal centre and splenic plasma cells in immunized mice is B cell-autonomous. Flow cytometric analysis of splenocytes from Eμ-tTA mice ($n = 6$) or littermate control ($n = 6$) at day 7 after immunization with NP-KLH. **a** Representative stainings with antibodies against B220 and CD138 markers to identify plasma cells (left) or NP-specific IgG1+ germinal centre B cells (left) defined as IgM− IgD− Gr1− CD138− B220+ IgG1-NP+. For the gating strategy, please see the Supplementary Fig. 9b. **b** Frequencies of germinal centre NP-specific isotype-switched B cells (IgM− IgD− Gr1− CD138− B220+ IgG1-NP+) and splenic plasma cells (B220low CD138+). Data are presented as box-and-whiskers diagram with median and interquartile range. **P ≤ 0.01, Eμ-tTA vs. WT, two-way ANOVA with Bonferroni correction for multiple comparisons. **c** Frequencies of total NP-specific IgG1-secreting cells in spleen as determined by ELISPOT assay. Each dot represents data from a single mouse as the mean of assays in replicate wells, with four mice in each group, and summarizes two independent experiments. Data are presented as scatter plots with mean. ***P < 0.001, Eμ-tTA vs. WT, two-sample Kolmogorov–Smirnov test

results demonstrate that Tet-transactivator-driven toxicity clearly depends on DNA binding as (i) rtTA-expressing mice were perfectly able to mount a humoral immune response in the absence of doxycycline, but lost this ability upon administration of doxycycline (Tet-On) triggering transactivator binding, while (ii) tTA-expressing animals regained the ability to generate antigen-specific B cells upon Dox administration (Tet-Off) (Fig. 5a, b). ELISPOT analysis corroborated these results and the findings from this test correlated well with the loss of GCs in spleens from Dox-treated rtTA mice and the formation of GC-like structures in Dox-treated tTA mice (Fig. 6).

We hypothesized that promiscuous DNA binding of the tTA may lead to the death of GC B cells that are genomically unstable because they are undergoing somatic hypermutation and affinity maturation. However, on day 7 post immunization, terminal transferase-mediated dUTP nick end-labelling (TUNEL) analysis, aiming to detect an increase in cell death in lymph follicles, actually failed to reveal statistically significant differences in the number of dying cells in splenic lymph follicles (Supplementary Fig. 8). Yet, it remained plausible that at that stage after immunization, dying transgene-positive cells were already cleared effectively by macrophages, and homeostasis was achieved by the expansion of transgene-negative cells. In order to test if cell death was indeed involved in the observed phenotype, we generated vav-tTA mice that co-expressed a BCL2 transgene, a potent inhibitor of apoptosis, placed also under the control of the *vav*-gene promoter. Most strikingly, upon immunization, we observed that double-transgenic mice were perfectly able to launch an immune response and to generate double-transgenic antigen-specific activated B cells, GCs and plasma cells. Furthermore, enforced expression of BCL2 could restore the ability to secrete antigen-specific IgG1 in Ta-expressing cells, as assessed by ELISPOT analysis (Fig. 7). This suggests that the DNA-bound Tet-transactivator-mediated induction of apoptotic cell death in antigen-stimulated B cells curtails humoral immunity in mice. Consistently, almost no TUNEL-positive cells were found in spleens from mice overexpressing BCL2 (Supplementary Fig. 8), in line with our initial hypothesis.

## Discussion

The Tet-On/Off system for conditional expression of protein-coding genes or shRNAs has become increasingly used in preclinical models of cancer research to implement temporal and reversible control of transgene expression[19, 35–40]. A number of studies, however, have employed this system also to investigate activation of adaptive immune responses, such as upon infection in vivo[42, 57, 58]. Such studies have revealed valuable insights into the dynamic regulation of the GC reaction. However, it appears that the Tet-On/Tet-Off system can have non-specific side effects on immune responses, such as the one documented here, and this complicates the interpretation of data in some of these studies.

Examining T cell-dependent B cell responses in two different Tet-Off transactivator mouse lines, the vav-tTA and Eμ-tTA mice, and the Tet-On CAG-rtTA3 line, alone or in combination with Tet-regulated GFP-reporters encoding one of two different shRNAs, targeting Renilla or Fire fly luciferase under the control of a conventional Tet-responsive P$_{TET}$, or the improved tight P$_{TET-Tight}$ promoter, we observed the near complete loss of GCs and antigen-specific antibody producing cell formation upon immunization. Using these reporter strains, we could reveal that the extent of impairment of B cell activation directly correlated with the Tet-transactivator expression levels, since the loss of B cell function was exacerbated in the eGFP-positive cells in DTtight mice, compared to the conventional DT mice. Accordingly, the counter-selection of tTA-expressing cells and the appearance of the GFP-negative cells was enhanced in DTtight animals. In accordance to our observation, Obst et al.[59] have reported a complete loss of rtTA-dependent chimeric MHCII expression in B cells when the rtTA transgene was driven by a strong recombinant promoter consisting of an Eα enhancer fused with the MHCII invariant chain (Ii), while transgene expression was well-tolerated in macrophages and dendritic cells. Of note, the same chimeric MHCII transgene has been reported to be efficiently expressed in B cells if directly driven by the described recombinant li-promoter in a constitutive manner[60].

Besides the loss of GC and plasma cells, we also observed reduced numbers of marginal zone B cells in tTA mice upon immunization with antigen, consistent with the finding that marginal zone B cells also participate in the antibody response to

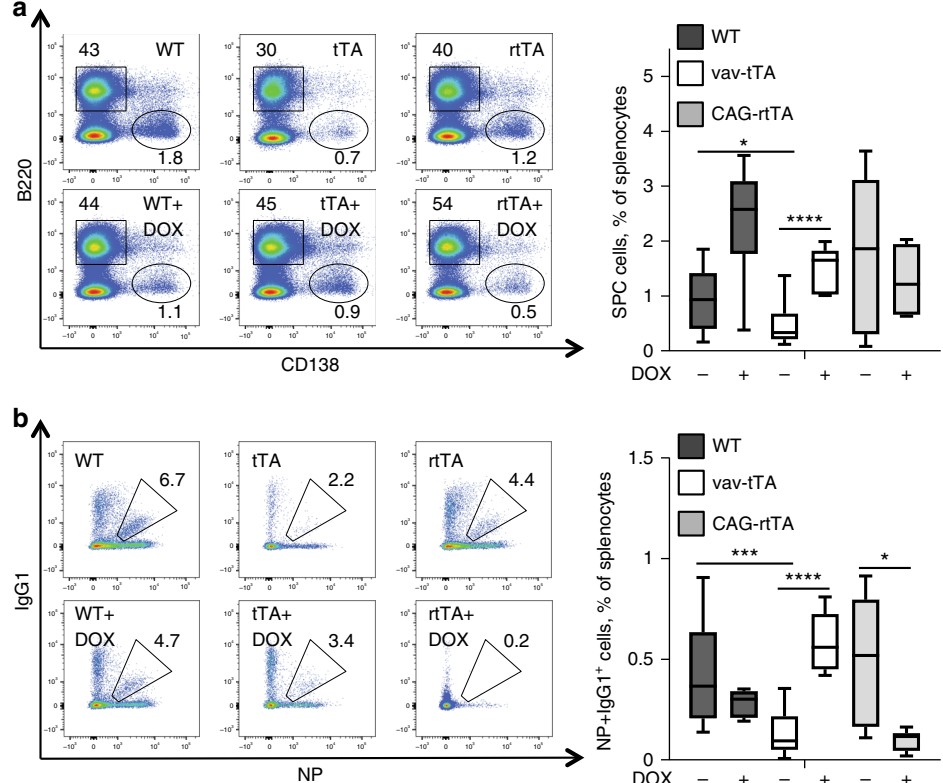

**Fig. 5** Tet transactivator-mediated cytotoxicity in antigen stimulated B cells depends on DNA binding. **a** Representative staining (left) and frequencies (right) of flow cytometric analysis of splenic plasma cells (B220$^{low}$ CD138$^+$) from vav-tTA ($n = 10$), CAG-rtTA ($n = 8$) or littermate control mice ($n = 9$) at day 7 after immunization with NP-KLH or vav-tTA mice ($n = 5$), CAG-rtTA mice ($n = 9$) or littermate control ($n = 8$) fed with doxycycline-containing food for 7 days prior and during the 7 days of NP-KLH challenge. **b** Representative FACS analysis (left) and frequencies (right) of germinal centre NP-specific isotype-switched B cells (IgM$^-$ IgD$^-$ Gr1$^-$ CD138$^-$ B220$^+$ IgG1-NP$^+$) from the mice of the indicated genotype as in **a**. For the gating strategy, please see the Supplementary Fig. 9b. Data are presented as box-and-whiskers diagram with median and interquartile range. *$P \leq 0.05$, **$P \leq 0.01$, ***$P < 0.001$, one-way ANOVA

T-dependent antigens[61, 62]. Along the same line, in tTA transactivator mice, we observed significant reduction in effector and effector memory T cells upon immunization and substantial impairment of the virus-specific cytotoxic T-cell response to LCMV infection. Interestingly, neither we, nor others, observed Tet-transactivator-mediated toxic effects during leukocyte development or in unchallenged mice at steady state[44, 46, 49–52, 63], with the exception of a modest reduction in the number of CD8$^+$ cDCs. However, others have already reported that antigen presentation is not impaired in rtTA-expressing DCs upon Dox treatment[64], therefore we anticipate that antigen presentation to T cells functions in vav-tTA mice as well. This indicates that presence of the tTA can be well-tolerated in tissues, even those with a high turnover rate, such as the blood. The escape from tTA/rtTA-mediated toxicity may be achieved by transgene silencing, as introduction of a GFP-reporter clearly demonstrated that tTA/rtTA expression is limited to a subset of blood cells, and varies strongly between tissues and is sometimes even variegated between littermates[44, 46, 65]. However, the tTA/rtTA-mediated toxicity becomes apparent in mice upon immunization or infection, or when B cells and T cells are subjected to an elevated genotoxic stress caused by the sustained proliferation and genome-remodelling processes.

For example, in Geraldes et al.[66], the authors make use of MMTV-tTA and TET-λLC double-transgenic mice on a knock-in μHC Rag1$^{-/-}$ background to control the expression of a transgenic B cell receptor (BCR). In those double-transgenic mice,

when the tTA is bound to DNA driving the expression of the BCR, an increase in the total number of B cells was noted, yet, follicular B cells were actually reduced when compared to wild type. Although the author did not discuss these data nor show an MMTV-tTA single transgenic mice control, this finding is compatible with tTA-mediated toxicity reported here. We foresee that the loss of follicular B cells in MMTV-tTA/TET-λLC/μHC/ Rag1$^{-/-}$ mice is driven by the heavy compensatory proliferation of B cells in the absence of Rag function.

While Tet-regulated RNAi or transgene expression did prove to be an extremely useful tool to research cancer development[19, 35–38, 39, 40], studies using this system for B and T cell activation in vivo should be interpreted with caution. For instance, Dominguez-Sola and colleagues used a Tet-On model where Omo-Myc, a selective MYC antagonist, was placed under the control of a tetracycline-inducible element, responsive to the rtTA transgene, driven from a CMV enhancer/β-actin promoter. Similar to what we observed in our animals expressing an irrelevant shRNA, Omo-Myc expression upon doxycline treatment caused a ~50% reduction in the percentages of GC B cells. The authors suggested that Omo-Myc expression disrupts MYC-driven transcriptional programs by abrogating its ability to bind and regulate canonical target genes, leading to cell-cycle arrest and apoptosis, therefore abrogating GC formation[57]. In the same study, the authors convincingly show that c-MYC is required for the initiation of the GC reaction, but is subsequently repressed by Bcl-6 in proliferating B cells that undergo affinity

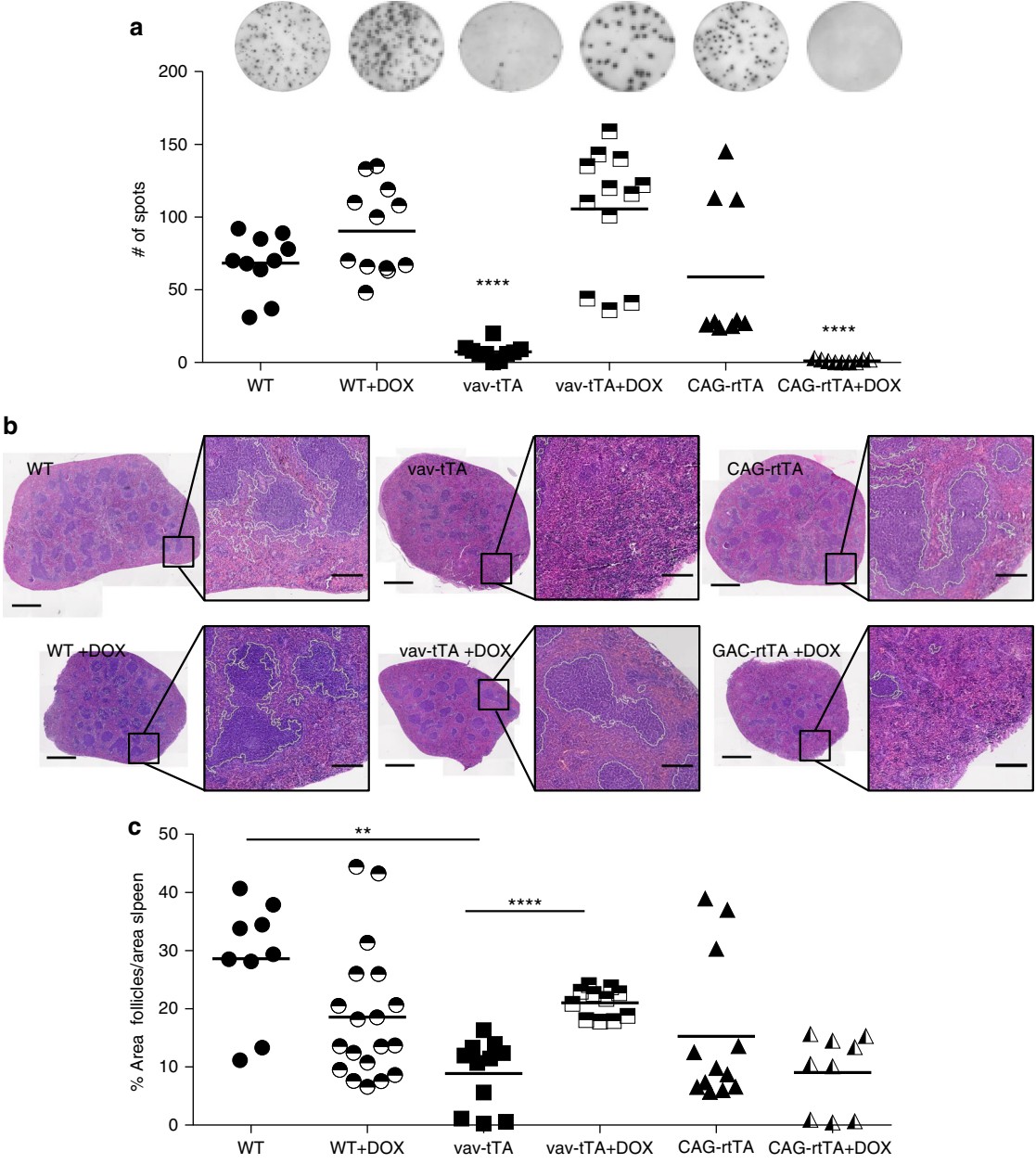

**Fig. 6** Tet transactivator-mediated cytotoxicity in antibody-secreting plasma cells depends on DNA binding. **a** Frequencies of total NP-specific IgG1-secreting cells in spleen as determined by ELISPOT assays on splenocytes from mice of the indicated genotypes at day 7 after immunization with NP-KLH or fed with doxycycline-containing food for 7 days prior and during the 7 days of NP-KLH challenge. Each dot represents data from a single mouse as mean of results from assays in replicate wells, with three to four mice in each group, and summarizes three independent experiments. Data are presented as scatter plots with mean. ****$P < 0.0001$, vs. WT or WT on DOX food, two-sample Kolmogorov–Smirnov test. **b** Representative H&E-stained splenic sections from the above-mentioned mice, scale bars represent 1 mm or 200 μm in the magnified views. White line demarcates the area used by the random forest classifier of Ilastik (see Methods section) for (**c**) quantification of the cumulative follicular area over total spleen section area. Data are presented as scatter plots with mean. **$P < 0.01$ vav-tTA vs. wild type and ****$P < 0.0001$, vav-tTA vs. vav-tTA on DOX-containing food, two-sample Kolmogorov–Smirnov test

maturation[57]. Therefore, it is puzzling how would induction of Omo-Myc on day 10 post immunization, after the GC have already formed, still diminished the number of GC B cells. These findings are equally consistent with rtTA-mediated toxicity upon doxycycline treatment, which may have escaped the attention of the authors, as single transgenic rtTA mice on doxycycline were not analysed in parallel[57].

Along similar lines, another group exploited the Tet system driving a reporter gene in order to track the inter-zonal movement of B cells within the GC in lymph nodes. In this study, the vav-tTA transactivator was used in combination with histone H2B-mCherry fusion protein expressed under control of the Tet-responsive promoter in B1-8$^{HI}$ mice, which contain an NP-specific immunoglobulin heavy chain transgene. The authors pulsed mice with doxycycline before immunization and observed that all GC B cells, in particular, the highly proliferating ones localized in the dark zone of the GC, were mCherry negative[42]. The authors assumed that, in those mice, NP-immunization and

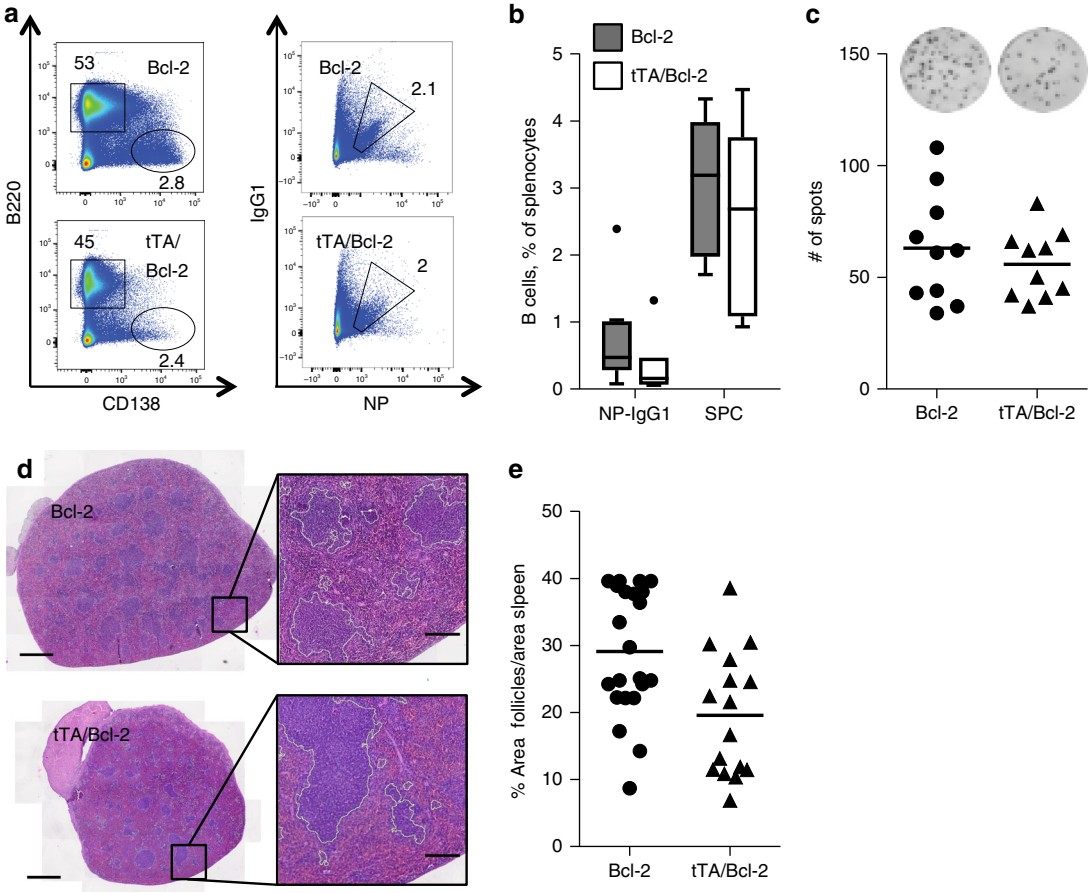

**Fig. 7** Overexpression of Bcl-2 prevents the transactivator-mediated cytotoxicity in antigen stimulated B cells. Flow cytometric analysis of splenocytes from vav-Bcl-2/vav-tTA double transgenic mice ($n = 10$) or vav-Bcl-2 littermate controls ($n = 10$) at day 7 after immunization with NP-KLH. **a** Representative staining using B220 and CD138-secific antibodies to identify plasma cells (left) and NP-specific IgG1+ germinal centre B cells as IgM− IgD− Gr1− CD138− B220+ IgG1-NP (right). For the gating strategy, please see the Supplementary Fig. 9b. **b** Frequency of GC and splenic plasma cells. Data are presented as box-and-whiskers diagram with median and interquartile range. **c** Frequencies of total NP-specific IgG1-secreting cells in spleens as determined by ELISPOT assays. Each dot represents data from a single mouse as mean of results from assays in replicate wells and five mice in each group, summarizing two independent experiments. Data are presented as scatter plot. **d** H&E-stained splenic sections from the above-mentioned mice, white line represent the follicle area used by the random forest classifier of Ilastik and **e** quantification of the cumulative follicular area over total spleen section area. Scale bars represent 1 mm or 200 μm in the magnified views. No significant differences were observed

doxycycline treatment-dependent loss of mCherry expression would reflect cellular division. However, our data suggest that a loss of tTA expressing B cells is actually due to cellular toxicity and selection of tTA-negative, and thus mCherry-negative, B cells in the dark zone of the GC.

So far, to our knowledge, only one study reported the use of the Tet-Off system to study T cell activation upon infection[67]. Tewari et al. compared the virus-specific CD8 responses of $Lck^{int}$ mice, on and off Doxycycline, upon infection with a recombinant vaccinia virus. Those $Lck^{int}$ mice harbour a CD2-driven rtTA plus an $Lck$ transgene under control of the conventional TET pro-moter, crossed into an $Lck^{-/-}$ background. The authors did not perform the experiments in CD2-rtTA single transgenic nor reported data regarding the anti-viral responses in wild-type mice, but only compared the responses of $Lck^{int}$ mice on/off Dox administration. Clearly, this comparison is not very informative, since the T cells off Docxycycline are de facto $Lck$-deficient and, therefore, cannot respond to TCR signalling at all[68]. From our data, we would anticipate a loss of vaccinia virus-specific T cells in $Lck^{int}$ mice on Doxycycline when compared with wild-type controls.

Cytotoxicity by tTA/rtTA overexpression may not be limited to activated B and T lymphocytes, but it also affects the myeloid lineage. In recent studies, we and others suggested a role for the anti-apoptotic A1/BFL1 in the survival of granulocytes, monocytic suppressor cells as well as cDCs, based on the use of mice expressing an mi-shRNA targeting A1 under control of the vav-tTA transgene[45, 69, 70]. However, as tTA expression was not traceable in these early studies, only bulk analyses were possible on different leukocyte subsets. In this study, we report that tTA expression per se compromises granulocyte survival, however, not to the same extent as in combination with a mi-shRNA targeting A1. However, this tTA-mediated toxicity must have contributed to the death observed in Tet-Off A1 knockdown mice. Similarly, we have now shown that tTA expression causes a reduction in CD8+ cDCs, contributing to the loss of cDCs observed in A1 knockdown mice[69], albeit cDC survival was also impaired in A1 knock out mice[71]. The data presented here raise questions about the significance of A1 for survival of both myeloid and lymphocytes.

Finally, the cytotoxic effects reported here also readily explain why Tet-On/Off conditional systems work well when applied to silence tumour suppressors, such as p53 or PTEN, as their knockdown provides a survival advantage under such stress-conditions[19, 20, 40]. While we do not want to refute the work published before using Tet-On/Tet-Off or similar models, here we

aim to sound a note of caution about the generalization of results from those models on immune responses, as adverse effects intrinsic to the system can confound the results obtained. Ultimately, each system is only as good as its most stringent control and to study complex processes in vivo, we need to further optimize the animal models we use. While the overall usefulness of the doxycycline-regulated system for conditional transgene expression, when controlled for properly, remains undisputed, data generated in the absence of such stringent controls needs to be interpreted with sincere caution.

## Methods

**Transgenic mice.** The generation and genotyping of vav-tTA, TRE-Ren, TRE$_{tight}$-Ren, TRE-Firefly, CAG-rtTA, Eμ-tTA and vav-Bcl2 mice, all maintained on a C57BL/6 genetic background, were previously described[19, 44, 72]. Double-transgenic animals were generated by intercrossing of single transgenic mice and genotyped by PCR. Doxycycline was administered in food pellets (625 mg/kg, Sniff) over a period of 14 days, that is, 7 days prior immunization, until organ collection on day 7 post immunization. No randomization or blinding was used in these experiments as all mice underwent identical treatment protocols. Animal experiments were performed using 9–12-week-old mice of both sexes (~1:1), bred and maintained at the animal facilities of the Walter and Eliza Hall Institute of Medical Research and the Biocenter at the Medical University Innsbruck. All animal procedures were approved by the The Walter and Eliza Hall Institute Animal Ethics Committee and the Biocenter Animal Ethics Committee (MWF-66.011/0007-II/3b/2014 and BMWF-66.011/0008-II/3b/2014).

**Immunization with NP-KLH.** Immunization of 7-week-old mice comprised a single intra-peritoneal injection of 100 μg 4-hydroxy-3-nitrophenylacetyl hapten coupled to keyhole limpet hemocyanin (NP-KLH) at a ratio of 21:1 and precipitated in alum, as described previously[73]. In details to precipitate the NP-KLH with alum, 100 μg of NP-KLH were diluted in 100 μl HEPES-buffered Eagles minimal essential medium per mouse to immunize. The same volume of 10% alum (AlK(SO$_4$)$_2$ × 12 H$_2$O) in water was then added to the NP-KLH mix. The pH was adjusted to 6.5 to allow the precipitation. After extensive washing with PBS, the pellet of NP-KLH was resuspended in PBS at a concentration of 1 mg/ml and 100 μl per mouse were injected intraperitoneally. Spleens were collected 7 days post immunization.

**Antibodies and flow cytometric analyses.** Single cell suspensions were stained using directly conjugated antibodies to surface markers (Supplementary Table 1), obtained from eBiosciences (San Diego, CA, USA), BD Biosciences (San Jose, CA, USA) or BioLegend (San Diego, CA, USA). For dendritic cell subset analysis, the spleens were digested in collagenase/DNaseI and purified on a Nycodenz gradient, according to the manufacturers recommendation. All antibodies not indicated as purchased from a commercial provider were purified and coupled to fluorochromes in-house at The Walter and Eliza Hall Institute. Antigen-specific B cells were identified by simultaneously binding NP coupled to phycoerythrin (PE)- and allophycocyanin (APC)-conjugated rat anti-mouse IgG1 (Supplementary Table 1) as shown in Supplementary Fig. 9b and as previously described[73].

**Enzyme-linked immuno spot assay.** The frequency of antibody-secreting cells in a cell population was determined as described[73]. Cells were incubated O/N at 37 °C on pre-coated 96-well MultiScreen-HA filter plates (Millipore) with 20 μg/ml NP-16-BSA. Spots were visualized with goat anti-mouse IgG1 antibodies conjugated to horseradish peroxidise (Supplementary Table 1) and colour was developed by the addition of the substrate 3-amino-9-ethyl carbazole (Sigma-Aldrich). Plates were washed extensively and spots were counted with an AID ELISpot reader system (Autoimmune Diagnostika).

**Viral infection and intracellular cytokine staining.** Infection was initiated by injecting $2 × 10^3$ PFU LCMV WE into the tail vein (i.v.) of 6-week-old mice. Spleens were collected 8 days post infection. Analysis of LCMV-specific CD8$^+$ T cells in spleens of infected mice was performed by tetramer staining using H-2Db/GP33 (KAVYNFATM), H-2Db/GP276 (SGVENPGGYCL) and H-2Db/NP396 (FQPQNGQFI), conjugated with streptavidin-APC (eBioscience) in combination with fluorochrome-conjugated antibodies (Supplementary Table 1). Splenocytes were stimulated in vitro with virus-specific peptides GP33 (KAVYNFATM), GP276 (SGVENPGGYCL) and NP396 (FQPQNGQFI), and stained for the intracellular cytokines (Supplementary Table 1) and analysed by flow cytometry[74].

**Cell proliferation and cell death assay.** Splenocytes were stimulated with 100 U/ml of mIL-2, 10 μg/ml mIL-4, 1 μg/ml mIL-5 (all Peprotech) and 2 μg/ml F(ab′)2 fragments goat anti-mouse IgM (Jackson ImmunoRЕsearch), 2 μg/ml hamster anti-mouse CD40 mAb (3/23, BD), or 100 nM ODN74 5′–AAAAAAAAAAAAAAACG

TTAAAAAAAAAAAA–3′ (Microsynth). Proliferation was assessed using the cell proliferation dye CPD eFluor 670 (eBioscience) and viability by 7-amino-actinomycin D (7AAD) or TO-PRO3 (Invitrogen) and Annexin-V exclusion and flow cytometric analysis.

**Histology.** Spleens were fixed in 4% PFA (paraformaldehyde) and embedded in paraffin. Three-micron sections were stained with haematoxylin and eosin stain for histological examination. Images were recorded using Olympus VS120 slide scanning, the images were then segmented and the splenic follicles were assessed using the object classification tool by Ilastik[75]. For simultaneous visualization of GCs and apoptotic cells, paraffin sections were deparaffinized and rehydrated. GCs were stained with fluorescein isothiocyanate (FITC)-conjugated peanut agglutinin after antigen retrieval in TE10/1, ph 8 at 80 °C for 30 min. Then the tissue sections were treated with 0.1% Triton X-100/0.1% sodium citrate buffer, and dehydrated by graded series of alcohol and chloroform, and apoptotic cells were detected by TUNEL. The TUNEL reaction was carried out in a humidified chamber for 1 h at 37 °C using recombinant terminal transferase and tetramethylrhodamine (TRITC)-dUTP (both Roche, Manheim, Germany) as described[76]. Nuclei were counterstained with DAPI (Sigma). Images were quantitatively analysed by a blinded observer using the image analysis software TissueQuest (TQ4.0, TissueGnostics, Vienna, Austria).

**Statistical analysis.** All data are presented as scatter plots or box-and-whiskers-diagram with median, the box length equals the interquartile range and circle represent the outliers, unless stated otherwise. The significance of differences among different groups was determined by two-way analysis of variance (ANOVA) with the Bonferroni correction for multiple comparisons or the Kolmogorov–Smirnov test or the two-tailed Mann–Whitney-U test for non-normally distributed data sets. The P values of < 0.05 were considered to indicate significant differences.

**Data availability.** The data that support the findings of this study are available from the authors on reasonable request.

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

## Acknowledgements

We are grateful to K. Rossi, C. Soratroi and I. Gaggl for excellent technical assistance or animal care. We thank R. Dickins, D. Tarlinton, J. Zuber and S. Lowe for mice or reagents. We are grateful to D. Bell for the help with the computational analysis. We thank V. Labi and M. Sochalska for the help with flow cytometric data analysis and cell sorting. This work was supported by the MCBO Doctoral College 'Molecular Cell Biology and Oncology' (W1101) and I-1298 (FOR2036), both funded by the Austrian Science Fund (FWF) and the 'Österreichische Krebshilfe Branch Tirol'. M.J.H. is supported by the National Health and Medical Research Council, Australia project grant APP1049720.

## Author contributions

E.O. performed experiments, analysed data and wrote the paper; V.P., K.S., E.C., R.S., M.P. and S.P. performed experiments and/or provided essential reagents and analysed data; M.J.H., A.S. and A.V. designed the research, analysed data, wrote the paper and conceived the study.

## Additional information

**Competing interests:** The authors declare no competing financial interests.

