## [Peer review file · Nature Communications]

Reviewers' comments:

Reviewer #1 (Remarks to the Author):

Villunger and colleagues report that mice bearing either the doxycycline-inducible (Tet-On) or doxycycline-repressible (Tet-Off) Tet-transactivators (tTAs) exhibit attenuated T or B cell specific immune responses that result in the apparent apoptosis of antigen-stimulated cells. Evidence is provided that while steady state levels of immune cells are unaffected, that the DNA binding of the tTA appears to drive apoptosis of activated lymphocytes. Conceptually, this manuscript provides important warning to investigators about the interpretation of data from commonly-used tTA-bearing mice, especially in the immune system. Overall, the data presented are strong and provide adequate support for the authors' conclusions.

1. Despite the strength of the presented data, there is little attempt made to address why expression of a tTA should lead to the loss of activated lymphocytes. Therefore, the story as currently presented is completely descriptive and lacks any assessment of how tTA DNA binding is promoting death of B and T cells. Any evidence of how tTA DNA binding can trigger this would enhance the impact of the study substantially.

Other weaknesses:

2. The ability to rescue B cell effects with ectopic BCL2 expression (Figure 7) are well performed, but no evidence is provided that the T cells can be rescued by anti-apoptotic expression.

3. The effect of tTA expression in B cells alone is shown to induce the repression of B cell immune responses and is used to argue the effects are B cell autonomous. No evidence is provided to demonstrate that T cells are also affected in a cell autonomous manner.

Reviewer #2 (Remarks to the Author):

This study by Ottina et al highlights important detrimental side-effects of tTA or rtTA-based systems for reversible inducible regulation of gene expression: DNA-binding of these fusion proteins imparts significant disadvantages (most likely cell death) upon antigen activated lymphocytes in vivo. The evidence presented in the manuscript is very strong. These experimental systems are very popular for the direct regulation of gene expression, for inducible RNA interference and for inducible CRISPR/Cas9-based systems. Therefore, the findings by Ottina et al should be presented to the general scientific public as soon as possible in a prominent manner. Finally, the authors also provide a remedy, namely expression of Bcl2, which seems to overcome the most important negative effects of tTA DNA binding, at least in the context of germinal center reactions and plasma cell formation. I have only some technical comments and general suggestions that do not detract from the overall importance of the message the authors want to convey.

Major points:

1) I am somewhat puzzled that the authors did not address whether the "survival of granulocytes, monocytic suppressor cells as well as conventional dendritic cells" is affected by tTA, as this directly impacts on their own published work, as they point out. No statements is given whether they are planning to re-assess or correct their publications.

2) One important aspect is missing in this study: What is causing the toxicity, DNA binding of the tetR protein or DNA binding of tetR fused to a VP16 transactivation domain (tTA, rtTA)? It remains possible that systems based on tetR alone do not suffer from these problems. While I do not think that the authors have to address this point experimentally in their present study, this aspect should be pointed out in the discussion at least.

Minor comments:

- 3) The Introduction could be focused somewhat more on doxycycline-inducible systems.
- 4) Fig 1a shows reduction of follicular B cells after immunization; In figure S1 there is no difference in follicular B cells in naïve mice. Is the difference due to the immunization? This is not adequately reflected in the text. Furthermore, in Figure 2b there is no difference in follicular B cells between GFP- and GFP+ DT cells.
- 5) I do not understand certain aspects in Figure 2: It appears only a fraction of DT splenocytes express GFP. It would be important to show here how many B cells express GFP. Figure 2 C and 2D show proportions of plasma cells and germinal center B cells in GFP- DT splenocytes comparable to wild-type mice (see Figure 1B, C). However, Figure 1B, C show severely reduced splenic germinal center and plasma cells in Vav-tTA mice compared to wild-type mice. Should GFP-negative DT (= Vav-tTA TRE-REN) germinal center and plasma cells, which appear to represent the majority of cells not resemble Vav-tTA splenocytes? As opposed to antigen-specific germinal center B cells (Figure 2D compared to 1C) there seems to be a reduction in antigen-specific plasma cells (Figure 2E compared to 1D) in GFP- DT cells compared to wild-type cells. I could only reconcile Figures 1 and 2 if GFP-negative cells were the clear minority of B cells in DT mice as suggested by Figure S2 – I therefore suspect that Figure 2A contains a mistake regarding the GFP+ cells? In line, Figure S3 shows clear effects also in GFP-negative FF cells compared to wild-type cells, with the exception of MZ B cells (S3C) and SPC (S3D); importantly, antigen-specific germinal center B cells and plasma cells are clearly reduced in GFP-negative cells. The authors have to analyze/interpret/describe these results more carefully.
- 6) Ideally, standard deviations should be shown instead of SEM. SEM reflect the confidence in the mean, while SD shows the variation in the data, which is more important to depict.
- 7) Figure legends S1, S2: how are the immune cell subsets defined?
- 8) Figure 2D: what is depicted here? GFP- and GFP+ cells? No legend is given.
- 9) Figure S2 shows changes in CD4 T cells, unlike stated in line 195 page 7.
- 10) I somewhat disagree with the last sentence in the manuscript: "Having said this, the overall usefulness of the doxycycline regulated system for conditional transgene expression remains undisputed by our findings." I would rather argue that their findings suggest that all studies that did not control for potential effects of tTA or rtTA expression alone have to be regarded with caution.

Reviewer #3 (Remarks to the Author):

Ottina et al. report that B and T cell responses are severely impaired in mouse lines carrying tetracycline transactivator transgenes. In vav-tTA transgenics, the differentiation of antibody secreting cells upon NP-KLH immunization and CD8+ T cell expansion and effector differentiation in response to LCMV are reduced due to cell-intrinsic toxicity of the transactivator. The findings are extended to the CAG-rtTA and Eμ-tTA transgenics and bolstered by data demonstrating that DNA binding of the transactivator is necessary for the effect.

The tet system has been used successfully in thymocytes^{1,2}, T cells^{1,3} and antigen-presenting cells^{4,5}, both in short-⁶ and long-term⁷⁻¹¹ experiments. The use of the tetracycline system in B cells, however, seems to be problematic as there is only one paper reporting success¹² and another one documenting the lack of B cell-specific rTA expression despite the use of a strong MHC

class II promoter⁵. Thus, while the documentation of cytotoxic effects of transactivator expression is certainly of interest to specialist readers, the generalization to lymphocytes contradicts the mentioned T cell work, of which none is mentioned in the ms. Since the paper also lacks novel information on lymphocyte biology as such submission to a more methods-oriented journal may be more appropriate.

1. Legname, G., Seddon, B., Lovatt, M., Tomlinson, P., Sarnier, N., Tolaini, M., Williams, K., Norton, T., Kioussis, D., and Zamoyska, R. (2000) Inducible expression of a p56Lck transgene reveals a central role for Lck in the differentiation of CD4 SP thymocytes. *Immunity* 12, 537-546
2. Saini, M., Sinclair, C., Marshall, D., Tolaini, M., Sakaguchi, S., and Seddon, B. (2010) Regulation of Zap70 expression during thymocyte development enables temporal separation of CD4 and CD8 repertoire selection at different signaling thresholds. *Sci Signal* 3, ra23
3. Labrecque, N., Whitfield, L. S., Obst, R., Waltzinger, C., Benoist, C., and Mathis, D. (2001) How much TCR does a T cell need? *Immunity* 15, 71-82
4. Jellison, E. R., Turner, M. J., Blair, D. A., Lingenheld, E. G., Zu, L., Puddington, L., and Lefrancois, L. (2012) Distinct mechanisms mediate naive and memory CD8 T-cell tolerance. *Proc Natl Acad Sci USA* 109, 21438-21443
5. Obst, R., van Santen, H. M., Mathis, D., and Benoist, C. (2005) Antigen persistence is required throughout the expansion phase of a CD4+ T cell response. *J Exp Med* 201, 1555-1565
6. Filby, A., Seddon, B., Kleczkowska, J., Salmond, R., Tomlinson, P., Smida, M., Lindquist, J. A., Schraven, B., and Zamoyska, R. (2007) Fyn regulates the duration of TCR engagement needed for commitment to effector function. *J Immunol* 179, 4635-4644
7. Leignadier, J., Hardy, M. P., Cloutier, M., Rooney, J., and Labrecque, N. (2008) Memory T-lymphocyte survival does not require T-cell receptor expression. *Proc Natl Acad Sci USA* 105, 20440-20445
8. Leignadier, J., Rooney, J., Daudelin, J. F., and Labrecque, N. (2011) Lowering TCR expression on naive CD8+ T cells does not affect memory T-cell differentiation. *Immunol Cell Biol* 89, 322-325
9. Schim van der Loeff, I., Hsu, L. Y., Saini, M., Weiss, A., and Seddon, B. (2014) Zap70 is essential for long-term survival of naive CD8 T cells. *J Immunol* 193, 2873-2880
10. Seddon, B., Tomlinson, P., and Zamoyska, R. (2003) Interleukin 7 and T cell receptor signals regulate homeostasis of CD4 memory cells. *Nat Immunol* 4, 680-686
11. Tewari, K., Walent, J., Svaren, J., Zamoyska, R., and Suresh, M. (2006) Differential requirement for Lck during primary and memory CD8+ T cell responses. *Proc Natl Acad Sci USA* 103, 16388-16393
12. Geraldès, P., Rebrovich, M., Herrmann, K., Wong, J., Jack, H. M., Wabl, M., and Cascalho, M. (2007) Ig heavy chain promotes mature B cell survival in the absence of light chain. *J Immunol* 179, 1659-1668

Point to point reply:

Reviewer #1 (Remarks to the Author):

Comment: Conceptually, this manuscript provides an important warning to investigators about the interpretation of data from commonly-used tTA-bearing mice, especially in the immune system. **Overall, the data presented are strong and provide adequate support for the authors' conclusions.**

Reply: We want to thank this referee for his/her overall very favorable review of our work.

Comment: Despite the strength of the presented data, there is little attempt made to address why expression of a tTA should lead to the loss of activated lymphocytes. Therefore, the story as currently presented is completely descriptive and lacks any assessment of how tTA DNA binding is promoting death of B and T cells. Any evidence of how tTA DNA binding can trigger this would enhance the impact of the study substantially.

Reply: We agree with this referee's opinion that elucidating the mechanism triggering toxicity upon tTA/rtTA binding to genomic DNA would constitute a significant advance in our study. In fact, we did actively try to answer this experimentally, e.g. by sorting transitional and follicular B cells from wt and vav-tTA mice and monitored their survival and proliferation in response to mitogen stimulation, using either anti-IgM F(ab)₂ fragments, a CD40 crosslinking antibody or a CpG-motive containing oligonucleotide *in vitro* (**Fig S4**). Yet, in this setting, we could not observe a significant impairment in proliferation or survival of B cells expressing the tTA transgene. However, it has to be pointed out that these results are not unprecedented since *ex vivo* stimulation of B cells cannot faithfully mimic the complex microenvironmental signals that control B cell proliferation, survival and differentiation in germinal centers (GC). Therefore, to obtain better insight, we decided to measure changes in cell death genes in GC B cells, isolated after immunization with NP-KLH. To overcome obvious problem associated with *in vivo* counter-selection and cell loss in vav-tTA mice, we decided to sort germinal center B cells from CAG-rtTA mice 7 days after immunization and to pulse them *ex vivo* with doxycycline for 2 or 4 hours prior collecting cells for RNA isolation. Unfortunately, we suffered repeat difficulties in isolating GC B cells of sufficient viability by cell sorting as these cells do not seem to withstand the sorting procedure well. This was reflected by cell death rates around 70% after two hours of *in vitro* culture, regardless of the absence or presence of doxycycline. As such, we believe that all expression data generated from RNA isolated from these cells would have been uninterpretable and stopped.

In conclusion, we do agree that the identification of the molecular mechanism that leads to tTA/rtTA mediated cell death in activated lymphocytes *in vivo* is of central importance to develop improved animal models and standardized treatment regimens to allow reliable studies of B and T cell activation using this system. We can only openly admit that we were falling short in addressing this issue in a satisfactory manner in the time given and with the resources available for revision.

Comment: The ability to rescue B cell effects with ectopic BCL2 expression (Figure 7) are well performed, but no evidence is provided that the T cells can be rescued by anti-apoptotic expression.

Reply: Indeed, we did not formally prove that the loss of virus-specific, activated T cells observed in tTA mice can be rescued by BCL2 overexpression. We now specify in the text that we were performing rescue experiments only activated B cells, limiting this conclusion to B cells. Result section, line 236.

Comment: The effect of tTA expression in B cells alone is shown to induce the repression of B cell immune responses and is used to argue the effects are B cell autonomous. No evidence is provided to demonstrate that T cells are also affected in a cell autonomous manner.

Reply: We appreciate this concern raised by this referee. Yet, Lck-tTA mice that would be needed to properly address this, are not available to us. The loss of virus-specific CD8 T cells upon acute LCMV-WE infection, however, should not be impaired due to impaired B cell activation during viral infection that play no role in primary anti-viral response, expansion or activation of virus-specific cytotoxic T lymphocytes. In support, it was shown that B cell deficient mice present normal percentages and numbers of CD8+ T cells during the initial response to LCMV infection (day 3-8 post infection), the time point we used in our analysis (Asano M.S. and Ahmed R. J. Exp. Med 1996; Di Rosa F and Matzinger P. J EXP Med 1996; Whitmire JK et al J. Immunol. 2009).

Likewise, the loss of virus-specific T cells is unlikely due to the mild reduction of CD8+ cDCs that we observed in vav-tTA mice (**Fig. S2**), since these DCs are dispensable for priming of naive T cells and generation of a memory response during acute LCMV infection (Hilpert C. et al. J Immunol. 2016). We now describe these facts in relation to the use of the LCMV model in the result section in more detail (lines 217-220) and hope that this will clarify this issue.

Reviewer #2 (Remarks to the Author):

Comment: This study by Ottina et al highlights important detrimental side-effects of tTA or rtTA-based systems for reversible inducible regulation of gene expression: DNA-binding of these fusion proteins imparts significant disadvantages (most likely cell death) upon antigen activated lymphocytes in vivo. The evidence presented in the manuscript is very strong. These experimental systems are very popular for the direct regulation of gene expression, for inducible RNA interference and for inducible CRISPR/Cas9-based systems. **Therefore, the findings by Ottina et al should be presented to the general scientific public as soon as possible in a prominent manner.** Finally, the authors also provide a remedy, namely expression of Bcl2, which seems to overcome the most important negative effects of tTA DNA binding, at least in the context of germinal center reactions and plasma cell formation.

I have only some technical comments and general suggestions that do not detract from the overall importance of the message the authors want to convey.

Reply: We want to thank this referee for his/her overall very positive and constructive review of our work. We appreciate the suggestions of this referee to improve the quality and the impact of our study.

Major points:

Comment: I am somewhat puzzled that the authors did not address whether the “survival of granulocytes, monocytic suppressor cells as well as conventional dendritic cells is affected by tTA, as this directly impacts on their own published work, as they point out. No statement is given whether they are planning to re-assess or correct their publications.

Reply: This is a valid argument. We agree that it is important to address the impact of tTA expression on other hematological compartments, also in light of previously published results by us and others. To address the critique of this referee, we now include also data on the composition of the myeloid compartment in tTA and DT mice in bone marrow and spleen (**Fig S5a-c**). We found a high degree of variability in reporter expression in myeloid cells, with rather low GFP expression in granulocytes. These findings are in line with our own former analyses in Vav-Venus reporter mice (Grespi et al, PlosOne 2011) and the GFP expression pattern reported by Dickins and colleagues, who also describe low Vav-gene promoter activity in the myeloid compartment (Takiguchi et al, PlosOne 2013), suggesting this promoter is less active in these cells.

Furthermore, we tested whether the survival of sorted granulocytes from vav-tTA and DT mice differed from wt cell and whilst the bulk of tTA-derived granulocytes did not show any survival impairment, GFP+ granulocytes presented higher rate of apoptosis compared to the wt controls ex vivo (**Fig. S5d**). This observation is of importance for our own previously published work, where we noted increased cell death in granulocytes from mice expressing an mi-shRNA targeting the survival protein A1, expressed under control of the Vav-gene promoter. It has now come clear that tTA-driven toxicity has confounded these results and explains why increased granulocyte cell death was only in one out of two model system used to knock down A1. This is now discussed in the results part (lines 175-182) and in the discussion (lines 378-381) of the current manuscript.

We were also able to monitor the number and composition of DCs in vav-tTA mice in steady state (**Fig. S2**). While the number and percentage of CD11c+ cells were comparable to what we found in wt animals, we observed a mild reduction in the percentages of CD8+ conventional DCs, paralleled by an reciprocal increase in CD11b+ cDCs. These data and their potential impact on previously published work DC survival are now described in detail in the result (lines 154-156) and in the discussion sections (lines 308-311 and lines 381-384) of the manuscript. We had however no means to address the impact of tTA expression on MDSC.

Comment: One important aspect is missing in this study: What is causing the toxicity, DNA binding of the tetR protein or DNA binding of tetR fused to a VP16 transactivation domain (tTA, rtTA)? It remains possible that systems based on tetR alone do not suffer from these problems. While I do not think that the authors have to address this point experimentally in their present study, this aspect should be pointed out in the discussion at least.

Reply: We do understand the reasoning behind this reviewer's suggestion. The TetR moiety confers sequence specific DNA binding to TetO sequences and sensitivity to tetracyclines but does not allow to induce transcriptional activation in mammalian cells (Gossen and Bujard PNAS 1992). The tTA we used here is a hybrid transcription factor resulting from the fusion of the prokaryotic Tet repressor (TetR) with a viral transcriptional transactivation domain, i.e. of HSV VP16. This transcriptional transactivation domain allows the recruitment of the transcriptional machinery to the promoter making the system functional in a eukaryotic environment. Hence, it might be beneficial to replace the VP16-domain in the tTA or rtTA with other, eukaryotic transcriptional transactivation domains, or an optimized variant of VP16 itself, to reduce toxicity. Along this line, the tTA2 system has been developed by the TET System company. The hybrid tTA2-TetR transactivator replace the original VP16 transcription activation domain of 127 amino acids [207-335

amino acid at the C-term] by so-called "acid domains" of only 13 amino acids [207-246 amino acid at the C-term]. Nevertheless to date the P_{TET-tight} and the silencer tTS are the most commonly used modifications adopted to reduce leakiness of the TET system (lines 118-128).

Minor comments:

Comment: The Introduction could be focused somewhat more on doxycycline-inducible systems.

Reply: We have now re-edited the introduction part and inserted a more detailed description of the Tet-system in lines: 118-123.

Comment: Fig 1a shows reduction of follicular B cells after immunization; In figure S1 there is no difference in follicular B cells in naïve mice. Is the difference due to the immunization? This is not adequately reflected in the text. Furthermore, in Figure 2b there is no difference in follicular B cells between GFP- and GFP+ DT cells.

Reply: This interpretation is correct. We indeed observed a reduction in FO B cells only upon immunization in tTA mice and in double-transgenic DT mice in GFP- and GFP+ subsets, with a greater loss in the latter. The same holds true for the loss of MZ B cells. We have now edited **Fig 2** presenting data from DT mice subdivided on the basis of GFP expression, next to data from wt controls, to allow easier evaluation of the data. We also edited the text to highlight the differences in GFP expression in DT mice in steady state and upon infection in lines 183-186.

Comment: I do not understand certain aspects in Figure 2: It appears only a fraction of DT splenocytes express GFP. It would be important to show here how many B cells express GFP. Figure 2C and 2D show proportions of plasma cells and germinal center B cells in GFP- DT splenocytes comparable to wild-type mice (see Figure 1B, C). However, Figure 1B, C show severely reduced splenic germinal center and plasma cells in Vav-tTA mice compared to wild-type mice. Should GFP-negative DT (= Vav-tTA TRE-REN) germinal center and plasma cells, which appear to represent the majority of cells not resemble Vav-tTA splenocytes? As opposed to antigen-specific germinal center B cells (Figure 2D compared to 1C) there seems to be a reduction in antigen-specific plasma cells (Figure 2E compared to 1D) in GFP- DT cells compared to wild-type cells. I could only reconcile Figures 1 and 2 if GFP-negative cells were the clear minority of B cells in DT mice as suggested by Figure S2 – I therefore suspect that Figure 2A contains a mistake regarding the GFP+ cells? In line, Figure S3 shows clear effects also in GFP-negative

FF cells compared to wild-type cells, with the exception of MZ B cells (S3C) and SPC (S3D); importantly, antigen-specific germinal center B cells and plasma cells are clearly reduced in GFP-negative cells. The authors have to analyze/interpret/describe these results more carefully.

Reply: We realized that our wording and presentation was not carefully enough describing all the different mouse lines and model systems used in our study. The reason of the different responses to immunization of the GFP+ and GFP- subsets in DT_Renilla (Ren) and DT_Firefly (FF) mice is due to differences in the TET-promoters which drive GFP-reporter expression. In DT_Ren mice (**Fig S6**), the GFP reporter is under control of the $P_{TET-tight}$ promoter, therefore, now referred to in the text and figures as “DTtight”. While in DT-FF mice (**Fig 2 and S2**) GFP is placed under the control of the conventional TET promoter (Premisrirut P et al. Cell 2011). We did repeat the immunization experiments and FACS analysis with mice harboring the GFP and mi-shRNA targeting Renilla construct under the control of the conventional TET promoter and those mice and DT-FF mice are now referred to as “DT” mice in the manuscript.

Both promoters consist of an array of seven tet operators (tetO) sequences upstream a TATA-box, however, while the conventional TET promoter has a CMV-minimal promoter element positioned downstream to the tetO sequences (**Fig. S3a**), the TET-tight promoter (**Fig. S6a**) consist only of the CMV sequence from position -35 to +12, i.e., minimal initiator sequence without downstream promoter element (DPE) by Clontech. We are now describing the use of the conventional TET-promoter and the TET-tight version in the introduction section (lines 118-123).

The TET-tight system was developed to reduce leakiness and increase the levels of transgene expression. In consequence, this system also depends on higher levels of the tTA/rtTA for transgene expression. We observed that expression is tight and high, but apparently variegated in the absence of a positive selection pressure, such as p53 knockdown, where the system was first described (Premisrirut et. al. Cell 2011). For this reason the GFP+ cells in DTtight mice are found to express more tTA compared to the GFP+ cells from conventional DT mice used, as judged based on qPCR analysis, **Fig. S3c-S6b**). This may also account for increased toxicity in the GC of DTRen mice vs. DTLuc mice vs. wt mice (mean of GC on total B cells: 0.0014% GFP+ DTtight ($P_{TET-Tight}$) vs. 0.16482% GFP+ DT (conventional P_{TET}) vs. 0.9256% wt).

As this reviewer points out, both DTtight and DT mice do exhibit variegation in GFP expression. We have now edited **Fig. 2**, where we show the GFP expression levels in total B cells, germinal center B cells, plasma cells and T

cells in reporter mice upon immunization (**Fig 2a**) and in **Fig S3b** the GFP levels in B and T cells in steady state. Indeed, the number of GFP-negative B cells increases upon immunization, correlating with tTA mediated toxicity upon immunization forcing counterselection.

It has been reported previously that heterogeneous transgene expression can be due to silencing of viral CMV promoter elements (Furth, P. A., et al *Nucleic Acids Res* 19, 6205-6208 (1991)), however, in our setting, we can envision that the GFP-negative subset of cells contains cells that escaped the tTA/rtTA mediated toxicity by silencing/reducing tTA expression (**Fig S3c and S6b**). Therefore, we needed to point out that GFP-negative cells, most likely, do not faithfully resemble either wt nor GFP-positive cells, but a mixed population of cells with different properties. We edited the text to better explain the mouse strains in relation to the phenotypes observed, in both the result (lines 167-168 and 194-201) and discussion section (line 285-286).

Comment: Ideally, standard deviations should be shown instead of SEM. SEM reflect the confidence in the mean, while SD shows the variation in the data, which is more important to depict.

Reply: We see merit in this argument and agree that a depiction of the variation in the data is critical. As we are not dealing with normally distributed data, we now show data as scatter plot or box and whisker plots, showing median and interquartile range, the outliers are shown as single points. We hope that this will provide a better visualization of our data, without irritating other the reviewers.

Comment: Figure legends S1, S2: how are the immune cell subsets defined?

Reply: We edited the figure legend: Percentages of transitional type 1 (T1) B cells (B220^{high} IgM^{high} CD23^{low} CD21^{low}), transitional type 2 (T2) B cells (B220^{high} IgM^{high} CD23⁺ CD21^{high}), follicular (FO) B cells (B220^{high} IgM^{low} CD23⁺ CD21^{low}) and marginal zone (MZ) B cells (B220^{high} IgM^{high} CD23^{low} CD21^{high}); Percentages of CD4⁻CD8⁻ double negative (DN), CD4⁺CD8⁺ double positive (DP) as well as CD4⁺CD8⁻ (CD4) and CD4⁻CD8⁺ (CD8) single positive thymocytes and double negative (DN) stages 1-4 (CD25⁻CD44⁺ DN1; CD25⁺CD44⁺ DN2; CD25⁺CD44⁻ DN3; CD25⁻CD44⁻ DN4); Frequencies of CD4⁺ and CD8⁺ T splenocytes and naïve (CD4⁺ CD62L^{high} CD44⁻), effector and effector memory (CD4⁺ CD62L⁻ CD44⁻) and central memory (CD4⁺ CD62L^{high} CD44⁺) CD4 T cells, or naïve (CD8⁺ CD62L^{high} CD44⁻), effector and effector memory (CD8⁺ CD62L⁻ CD44⁻) and central memory (CD8⁺ CD62L^{high} CD44⁺) CD8 T cells.

Comment: Figure 2D: what is depicted here? GFP- and GFP+ cells? No legend is given.

Reply: We have added the figure legend to each single plot.

Comment: Figure S2 shows changes in CD4 T cells, unlike stated in line 195 page 7.

Reply: We have changed notable to prominent line 213.

Comment: I somewhat disagree with the last sentence in the manuscript: "Having said this, the overall usefulness of the doxycycline regulated system for conditional transgene expression, when controlled for properly, remains undisputed by our findings." I would rather argue that their findings suggest that all studies that did not control for potential effects of tTA or rtTA expression alone have to be regarded with caution.

Reply: We appreciate this encouragement for a stronger statement and have changed this, with pleasure, lines 394-95.

Referee #3

Comment: The tet system has been used successfully in thymocytes^{1,2}, T cells^{1,3} and antigen-presenting cells^{4,5}, both in short-⁶ and long-term⁷⁻¹¹ experiments. The use of the tetracycline system in B cells, however, seems to be problematic as there is only one paper reporting success¹² and another one documenting the lack of B cell-specific rtTA expression despite the use of a strong MHC class II promoter⁵. Thus, while the documentation of cytotoxic effects of transactivator expression is certainly of interest to specialist readers, the generalization to lymphocytes contradicts the mentioned T cell work, of which none is mentioned in the ms. **Since the paper also lacks novel information on lymphocyte biology as such submission to a more methods-oriented journal may be more appropriate.**

Reply: We thank this reviewer for valuable comments, yet, with all due respect we disagree with the idea that our study is not suitable for Nature Communication, as it does not provide new insights into lymphocyte development.

Actually, we believe a general journal, such as Nat. Communications, is well suited, as the impact of our findings on adaptive immunity using the TET system calls for a careful revalidation & re-interpretation of a number of

published findings and, secondly, go beyond the hematopoietic system. Indeed, we are convinced that our work is of general interest and relevant across several disciplines as it highlights major limitations of the almost ubiquitously used TET system for conditional transgene expression. We do believe that our finding need to be shared with the scientific community in the broadest possible way, making Nature Communication the ideal medium to spread this information among a multidisciplinary readership.

However, we do really appreciate the detailed bibliography that this reviewer brought to our attention, as some of these studies that we were unaware of actually support our observations while no single one is at odds with our findings. Hence, we hope that this referee sees enough merit in the revised version of our manuscript and supports its publication.

For example, in Geraldès et al, (ref 12) the authors make use of MMTV-tTA and TET- λ LC double-transgenic mice on a knock-in μ HC Rag1^{-/-} background to control the expression of a transgenic B cell receptor (BCR). It is clearly visible in figure 2A (fourth row) that in double-transgenic mice OFF-DOX (tTA bound to DNA; BCR expressed) there is a reduction of FO B cells (B220⁺ CD21^{low}CD23⁺) that correlates with a relative increase of marginal zone B cells (B220⁺ CD21⁺CD23^{low}), compared to wild type. Although the author did not discuss these data nor show an MMTV-tTA single transgenic control or trace transgene expression on a cellular level, this finding is compatible tTA-mediated toxicity. We foresee that the loss of follicular B cells in those of MMTV-tTA/TET- λ LC/ μ HC/Rag1^{-/-} mice, similar to what we observed in vav-tTA mice upon immunization (Fig. 1), is driven in those settings by the heavy proliferation the B cells have to go through to fill the niche in Rag^{-/-} mice. We now include this observation in our discussion on page 11.

In line with our finding, Obst et al, (ref 5), showed that whilst rtTA-mediated transgene expression upon DOX addition is efficient in DCs and macrophages, it is very low in B cells. Those mice have a rtTA transgene driven by a strong recombinant promoter, consisting of an E α enhancer fused with the MHCII invariant-chain (Ii) promoter. Of note, this recombinant promoter was previously reported to drive a very strong expression of another transgene, encoding a chimeric Ii cDNA with a segment of moth cytochrome c (MCC) instead of the CLIP region, as reporter in FO B cells (van Santen HM, et al. JIM 2000). Since critical controls that would exclude toxicity in B cells in the Obst paper are missing, we can only propose that the absence of rtTA expressing B cells upon DOX treatment could be due to counter-selection upon the rtTA binding to DNA. We are now discussing these papers, in the discussion section on page 10 lines 293-299. Moreover, also Jellison, E. R., et al (ref 4) clearly showed that antigen presentation was not impaired in rtTA-expressing DCs upon DOX treatment, in agreement with our findings that vav-tTA mice show no reduction in overall DC numbers in the spleen. Hence,

antigen-presentation to T cells should work normally in these mice. We are now citing this paper in line 310 of the text.

We can confirm that, as point out by this reviewer, tTA or rtTA expression in the presence or absence of DOX is generally well-tolerated in immature thymocytes or mature T cells in steady state/homeostatic conditions (ref 1, 2, 3, 9, Ottina E. et al. Blood 2012; Sochalska, M et al. Cell death and differentiation 2015; Takiguchi, M. et al. PlosOne 2013). In support of this notion, we are now presenting data regarding the thymocyte subset composition in vav-tTA mice in Fig. S3 of the revised manuscript. In addition, as we reported previously (Ottina E. et al. Blood 2012) and as now shown in Fig. S4, we did not observe gross impairment in survival or proliferation of tTA expressing CD4 and CD8 T cells upon stimulation *in vitro*, in accordance to ref 6 and 7. We now also refer to this fact in lines 171-173 and 307 of the manuscript.

Tewari et al. (Ref 11) compared the virus specific CD8-responses of the Lck^{int} mice, ON and OFF DOX, upon infection with a recombinant vaccinia virus. Lck^{int} mice harbor a CD2-driven rtTA and Lck transgene under control of the conventional TET-promoter on an Lck^{-/-} background. In this case a comparison between infected Lck^{int} and wt mice was not presented but only the responses ON and OFF DOX. However, this comparison is not very informative, since the T cells OFF DOX are essentially Lck-deficient cells and, therefore, cannot respond to TCR signaling, exhibiting a completely impaired proliferative responses (see also Trobridge et al Eur.J. Immunol. 2001), which is a much more severe loss of T cell function, compared to what we observed in our tTA mice upon viral infection (Fig. 3). From our data, we would predict a loss of vaccinia virus-specific T cells in Lck^{int} mice ON DOX when compared to wild type mice. The implications of our findings in relation to this study is now discussed on page 12-13, line 360-371

1. Legname, G., Seddon, B., Lovatt, M., Tomlinson, P., Sarner, N., Tolaini, M., Williams, K., Norton, T., Kioussis, D., and Zamoyska, R. (2000) Inducible expression of a p56Lck transgene reveals a central role for Lck in the differentiation of CD4 SP thymocytes. Immunity 12, 537-546
2. Saini, M., Sinclair, C., Marshall, D., Tolaini, M., Sakaguchi, S., and Seddon, B. (2010) Regulation of Zap70 expression during thymocyte development enables temporal separation of CD4 and CD8 repertoire selection at different signaling thresholds. Sci Signal 3, ra23
3. Labrecque, N., Whitfield, L. S., Obst, R., Waltzinger, C., Benoist, C., and Mathis, D. (2001) How much TCR does a T cell need? Immunity 15, 71-82
4. Jellison, E. R., Turner, M. J., Blair, D. A., Lingenheld, E. G., Zu, L., Puddington, L., and Lefrancois, L. (2012) Distinct mechanisms mediate naive and memory CD8 T-cell tolerance. Proc Natl Acad Sci USA 109, 21438-21443

5. Obst, R., van Santen, H. M., Mathis, D., and Benoist, C. (2005) Antigen persistence is required throughout the expansion phase of a CD4+ T cell response. *J Exp Med* 201, 1555-1565
6. Filby, A., Seddon, B., Kleczkowska, J., Salmond, R., Tomlinson, P., Smida, M., Lindquist, J. A., Schraven, B., and Zamoyska, R. (2007) Fyn regulates the duration of TCR engagement needed for commitment to effector function. *J Immunol* 179, 4635-4644
7. Leignadier, J., Hardy, M. P., Cloutier, M., Rooney, J., and Labrecque, N. (2008) Memory T-lymphocyte survival does not require T-cell receptor expression. *Proc Natl Acad Sci USA* 105, 20440-20445
8. Leignadier, J., Rooney, J., Daudelin, J. F., and Labrecque, N. (2011) Lowering TCR expression on naive CD8+ T cells does not affect memory T-cell differentiation. *Immunol Cell Biol* 89, 322-325
9. Schim van der Loeff, I., Hsu, L. Y., Saini, M., Weiss, A., and Seddon, B. (2014) Zap70 is essential for long-term survival of naive CD8 T cells. *J Immunol* 193, 2873-2880
10. Seddon, B., Tomlinson, P., and Zamoyska, R. (2003) Interleukin 7 and T cell receptor signals regulate homeostasis of CD4 memory cells. *Nat Immunol* 4, 680-686
11. Tewari, K., Walent, J., Svaren, J., Zamoyska, R., and Suresh, M. (2006) Differential requirement for Lck during primary and memory CD8+ T cell responses. *Proc Natl Acad Sci USA* 103, 16388-16393
12. Geraldès, P., Rebrovich, M., Herrmann, K., Wong, J., Jack, H. M., Wabl, M., and Cascalho, M. (2007) Ig heavy chain promotes mature B cell survival in the absence of light chain. *J Immunol* 179, 1659-1668

REVIEWERS' COMMENTS:

Reviewer #1 (Remarks to the Author):

The revised version of this manuscript is acceptable.

I strongly feel that this manuscript has sufficient merit and needs to be distributed to the research community as a cautionary tale about the use of tTA-bearing mice.

While it would be nice to understand the mechanism behind the toxicity, I understand that technical challenges have limited the authors' attempts.

Reviewer #2 (Remarks to the Author):

The authors have adequately addressed my few technical comments. I remain convinced that this careful and important study should be published prominently, so that a broad readership is alerted to the issue. Therefore I strongly recommend publication.

Reviewer #3 (Remarks to the Author):

The authors have sufficiently taken our comments to the initial version of the MS into consideration. Our earlier concerns exclusively related to the scope of the MS rather than to its technical merit. These remain, however, it is certainly an editorial decision whether the MS fits into the scope of Nature Communications.